# Body Remains Left by Bird Predators as a Reliable Source for Population Genetic Studies in the Great Capricorn Beetle *Cerambyx cerdo*, a Veteran Oak Specialist

**DOI:** 10.3390/insects12070574

**Published:** 2021-06-23

**Authors:** Aleksander J. Redlarski, Tomasz Klejdysz, Marcin Kadej, Katarzyna Meyza, Cristina Vasilița, Andrzej Oleksa

**Affiliations:** 1Department of Genetics, Faculty of Biological Sciences, Kazimierz Wielki University, Powstańców Wielkopolskich 10, 85-090 Bydgoszcz, Poland; kasiakow@ukw.edu.pl; 2Institute of Plant Protection-National Research Institute, Wegorka 20, 60-318 Poznan, Poland; t.klejdysz@iorpib.poznan.pl; 3Department of Invertebrate Biology, Evolution and Conservation, Faculty of Biological Sciences, University of Wrocław, Przybyszewskiego 65, 51-148 Wrocław, Poland; marcin.kadej@uwr.edu.pl; 4Research Group in Invertebrate Diversity and Phylogenetics, Faculty of Biology, Alexandru Ioan Cuza University in Iași, 700505 Iași, Romania; vasilita.cris@gmail.com

**Keywords:** DNA extraction, genetic analysis, microsatellite, SSR, non-invasive sampling

## Abstract

**Simple Summary:**

Molecular genetic techniques can support species conservation by providing information about processes critical to population survival. Unfortunately, obtaining biological material for DNA extraction is often associated with an adverse impact on the animals under study. In legally protected or threatened species, non-invasive sampling (i.e., sampling without injuring or disturbing an animal) is preferred as it carries no risk to the population’s survival. Here, we tested the possibility of using body remains left by bird predators for microsatellite genotyping in *Cerambyx cerdo*, a veteran oak specialist. We compared results obtained from such remains samples with samples of fresh beetle tarsi (i.e., invasive and destructive but non-lethal samples). We found that the sample type had no significant effect on PCR amplification efficiency; instead, it was strongly affected by allele length and individual heterozygosity. Allele frequencies were perfectly correlated for both sample types (*R*^2^ = 0.94). Although point estimates of individual inbreeding were higher in remains than fresh samples (medians 0.08 vs. 0.02, respectively), both groups were not significantly different from each other and zero. Our study demonstrated that non-invasive remains samples could provide satisfactory data for population–genetic studies. However, we highlight the problem of potentially biased inbreeding estimates, which may result from allelic dropout.

**Abstract:**

Obtaining biological material for DNA extraction is often challenging in organisms of conservation interest. Non-invasive sampling (i.e., sampling without injuring or disturbing an animal) is preferred as it carries no risk to the population’s survival. Here, we tested the possibility of using the body remains left by bird predators for microsatellite genotyping in *Cerambyx cerdo*, a veteran oak specialist. We compared results obtained from such potentially degraded samples with samples of fresh beetle tarsi (i.e., invasive and destructive but non-lethal samples). Using 10 SSR loci, we genotyped 28 fresh, and 28 remains samples. The analysis indicated that PCR amplification efficiency was not influenced by sample type but allele length and individual heterozygosity. Allele frequencies were perfectly correlated for both sample types (*R*^2^ = 0.94). Additionally, null allele frequencies and genotyping failure rates were not significantly different from zero. Although the point estimates of individual inbreeding rates (*f_i_*) were higher in remains than fresh samples (medians 0.08 vs. 0.02, respectively), both groups were not significantly different from each other and zero. Our study demonstrated that non-invasive remains samples could provide satisfactory data for population–genetic studies. However, we highlight the problem of biased inbreeding estimates, which may result from samples affected by allelic dropout.

## 1. Introduction

Molecular genetic techniques provide data for many types of evolutionary and ecological analyses, such as, among others, the resolution of evolutionarily significant units, sex identification, assessments of inbreeding, migration and estimation of the census and effective population sizes [1,2,3]. Although such information may be crucial for managing populations under conservation concern, a severe challenge for researchers wishing to apply molecular techniques is to provide adequate biological material for DNA extractions. To allow genetic marker amplification, DNA cannot be excessively degraded and should not contain PCR inhibitors [4]. Fresh material, taken directly from living individuals and preserved until extraction at low temperature and in protective liquids, is considered the best source of high-quality DNA. However, sampling from living individuals may be either destructive (lethal) or non-destructive (non-lethal). In the first case, the animal is killed to obtain the tissues necessary for genetic analysis. The second case usually requires an animal to be captured, and a biopsy or blood sample is taken invasively [5,6].

In legally protected or threatened species, destructive sampling should be avoided to reduce the impact of the data collection process on species survival. Even if the collection of tissue for research is non-lethal, it could still have an adverse impact on the fitness, behaviour or welfare of the individuals being tested. Hence, using various types of remains or secretions that can be collected without having to catch or disturb the animal as a DNA source is an appealing approach. This type of DNA collection is defined as non-invasive [5,6].

Particularly, in insect molecular studies, the collection of good quality samples often involves the lethal sampling of whole specimens. Most studies carried out without sacrificing individuals use invasive and disruptive but non-lethal methods such as wing clipping [7,8,9] or the amputation of tarsi [8,9,10,11,12,13,14] or antenna [15]. Non-invasive methods have been used less frequently because they often make it difficult to obtain a good quality and quantity of DNA extracts. Among others, DNA extracted from eggshells, larval and pupal exuviae and faeces [9,16,17,18,19,20] were successfully used for PCR amplification of relatively short barcoding sequences in different taxonomic groups of insects, including cicadas, dipterans, hymenopteran and beetles. DNA extracted from exuviae, or dead imagines were used to amplify microsatellites in dragonflies [21]. Rusterholz et al. [22] demonstrated that degraded remnants of traffic-killed beetle specimens could serve as a source of high-quality genomic DNA. All publications cited above evaluated the quality of extracted DNA by measuring purity and concentration or checking whether it was possible to amplify specific DNA fragments (usually, mtDNA barcode such as COI) in PCR reaction. None of the studies assessed the possibility of genotyping errors, which may occur if a DNA template with a high degree of degradation or low concentration is used in the PCR reaction [6]. Issues such as allele dropout can have severe consequences for population genetic inference, mainly when the analysis uses codominant nuclear markers based on a fragment length polymorphism (i.e., microsatellites or SSRs).

In this study, we focused on the non-destructive sources of DNA for genetic studies in the Great Capricorn beetle (*Cerambyx cerdo*, Linnaeus, 1758). *C. cerdo* is an important umbrella species that is protected under the EU’s Habitats Directive and threatened in many European countries [23]. It is also considered as an “ecosystem engineer”, creating the micro-niches both for invertebrates [24] and vertebrates [25,26]. For adequate protection of the species, it is necessary to better recognise the genetic population parameters as well as dispersal patterns. Non-destructive or non-invasive methods for collecting genetic material have not been so far developed for *C. cerdo*. As with other insects, non-lethal (but not non-invasive) amputation of one tarsus can be used [11]. Although it is possible to obtain good quality samples in this way, it is difficult to estimate how such treatment may affect an individual’s survival in the environment. Under natural conditions, mutilated beetles (e.g., with cut off fragments of antennas or legs due to interaction with other individuals of the same species or predators) could often be observed, and such injuries do not appear to cause significant impairment of individuals. Experiments also confirm that the loss of a small portion of the antenna may have minimal impacts on survivorship and reproduction in other insects [15,27]. Nevertheless, when studying threatened species, it is undesirable to cause additional risk factors for the population, especially those local and small.

The concept of non-lethal and non-invasive sampling could be extended to the remains of individuals left by predators. The great Capricorn beetle is relatively often the object of bird predation [28]. Especially if the species appears exceedingly abundantly in a given field site, birds (mainly the hooded crow *Corvus cornix*, own observation) seem to specialise in using this easily accessible food source. As the body of *C. cerdo* is heavily sclerotised (especially the head with antennas, elytrae, and legs), the birds limit their consumption to more convenient body parts (abdomen) and leave many residues after their feast. Such remains accumulate under trees, and conservation biologists may be tempted to use them for molecular research. Unfortunately, so far, there is no assessment of the possibility of using them to obtain DNA extracts that are good enough to be used in genotyping applications. Thus, in this study, we explore (1) whether DNA can be isolated from the remains of *C. cerdo* to allow the amplification of microsatellite markers in a PCR reaction, (2) what is the level of genotyping errors in fresh and remains samples, and (3) whether remnants of predated individuals are a good, random representation of the entire local population.

## 2. Materials and Methods

We conducted the study in the Middle Odra Valley in western Poland (Figure 1a). This area is one of the most important places of *C. cerdo* occurrence in Poland [29,30]. The species inhabits mainly veteran oaks growing on the Odra embankments, in roadside avenues and woodland pastures [31]. Two special areas of conservation (SACs) within the European Natura 2000 network were established: Kozioróg w Czernej (PLH020100) and Nowosolska Dolina Odry (PLH080014). Both SACs aim to protect habitats of *C. cerdo* in cultural landscapes and fragments of semi-natural alluvial forests along the river (habitat 91E0: alluvial forests with willow, poplar, alder, and ash—*Salicetum albo-fragilis*, *Populetum albae*, *Alnenion glutinoso-incanae*; habitat 91F0: alluvial forest with oak, elm, and ash—*Ficario-Ulmetum*).

We collected samples from 8 June to 7 July 2018 between the city Nowa Sól in the north-west (51°48′02″ N, 15°44′26″ E) and the village Czerna in the south-east (51°42′25″ N, 15°54′31″ E) (section of the Odra valley with a length of approximately 15 km). Since *C. cerdo* is a fairly good flier [32,33], it can be assumed that a single population inhabits the whole area (confirmed by our analyses presented later in this work).

Fresh samples (beetles’ tarsi) were amputated from beetles that we collected on oak trunks during the day and insects encountered during night searches with a flashlight. Several beetles were lured using the bait traps with cheap red wine, made of 5 L plastic bottles by cutting off the neck and placing it upside down on top of the bottle’s lower part to form a funnel (for trap design, see [34,35]). Thanks to the installation of a perforated partition, the beetles did not drown in the lure and were released after tarsi amputation. Fresh samples were immediately preserved in ethanol and transported in a portable cool box to the refrigerator as soon as possible. Regarding the remains of beetles, we collected body fragments lying under veteran oaks (Figure 1b). We suppose that a large part of them were the remains of insects killed by birds, although we cannot exclude other causes of mortality as well. To avoid collecting different fragments of the same individual and to control the total number of individuals sampled, we collected only heads with antennas. However, for DNA extraction, we used only two to three antenna segments, depending on their size, to obtain a similar tissue amount as for tarsi. Regardless of origin, the samples were immediately preserved in 90% ethanol and, after bringing them to the laboratory, were stored in the refrigerator (−20 °C) until DNA extraction, which we performed using the Insect DNA Kit (Omega Bio-Tek, Norcross, GA, USA) following the manufacturer’s protocol.

To characterise genetic diversity, we used 10 microsatellite loci developed for C. cerdo [36]. All of them were amplified in a single multiplex reaction, which was performed using the Multiplex PCR Kit (Qiagen, Hilden, Germany) according to the protocol recommended by the manufacturer in a volume of 10 μL (5 μL of 2 × QIAGEN Multiplex Master Mix, from 75 to 125 nM of each primer pair, 0.5 mg/mL of BSA, 10 ng of template DNA and ddH20 up to the total volume of 10 μL). The thermal PCR conditions were as follows: an initial incubation at 95 °C for 15 min; nine touchdown cycles: 94 °C for 30 s, 56 °C (−0.5 °C per cycle) for 1 min 30 s and 72 °C for 1 min; 24 cycles of 94 °C for 30 s, 52 °C for 1 min 30 s and 72 °C for 1 min; final elongation at 72 °C for 10 min. For amplification, we used PTC200 thermal cycler (MJ Research, Waltham, MA, USA). Separation of the amplified fragments was performed on an ABI PRISM 3130xl automated sequencer (Applied Biosystems, Foster City, CA, USA) using a LIZ 600 size marker (Applied Biosystems, Foster City CA, USA). The electropherograms were interpreted using the GeneMarker ver. 2.6.3 (SoftGenetics, State College, PA, USA). When amplification of one or more loci failed, we repeated the multiplex PCR for this particular sample.

To compare the differences between proportions of successful amplification for the two types of samples (i.e., fresh tarsi and remnants of predated individuals), we used the *Z*-test. We characterised each group using basic genetic diversity parameters, including the observed number of alleles (*N_a_*), observed heterozygosity (*H_o_*), expected heterozygosity (*H_e_*), and inbreeding coefficient (*F_is_*, which is calculated as 1-*H_o_*/*H_e_*), calculated in INEst ver 2 [37]. We used the same software to assess the frequencies of null alleles (*n*), random genotyping failure (*b*) and to provide unbiased multi-locus estimates of inbreeding coefficients (*F_i_*). We tested differences between fresh and remnant samples in the statistics listed above using Wilcoxon signed-rank tests for paired data. To assess the fit of the remains to fresh samples allele frequencies (corrected for the presence of null alleles using INEst), we used a linear hypothesis test assuming intercept = 0 and slope = 1, computed in ‘car’ ver. 3.0.10 [38] in R.

We tested differentiation between both types of samples using pairwise *F_st_*, according to Nei [39]. To check whether the obtained data sets will equally recognise the population’s genetic structure, we applied a Bayesian clustering method implemented in STRUCTURE [40]. The number of subpopulations (*K*) was determined using the Δ*K* approach [41].

Spatial genetic structure (SGS) was assessed by using a multi-locus kinship coefficient. The analyses were performed separately for both types of samples using the SPAGeDi ver. 1.3 software [42,43]. Kinship was estimated according to Nason’s formula [43]. Correlograms were obtained by averaging kinship coefficients within assumed distance classes, each containing an approximately equal (from 35 to 39) number of pairs. The number of distance classes (10) was selected using the Sturges rule. Standard errors were estimated by the jackknife procedure over loci.

To further investigate the influence of sample type on amplification efficiency, PCR products (peaks in electropherograms) were described with three variables—peak height, peak area, and peak quality score (Figure 2)—calculated with GeneMarker ver. 2.6.3 (SoftGenetics, State College, PA, USA). The latter characteristic represented a signal-to-noise ratio and peak shape, with lower values indicating poorer quality peaks. The variables were log or log + 1 transformed to approximate normal distribution. To account for collinearity, the first principal component (PCA) from height, area and score was used as the single measure of amplification efficiency in subsequent analyses. To assess the effect of DNA source (i.e., fresh vs. remnant) on amplification efficiency, we built linear mixed-effects models using the ‘lmer’ function in lme4 ver. 1.1.26 [44] in R ver. 4.0.4 [45]. As we expected that the allele fragment length (numeric variable, in bp) and heterozygosity of an individual at a given locus (binary factor) could have a systematic and predictable impact on our dependent variables, we used them as additional fixed effects (apart from the effect of DNA source). In contrast, locus (factor, with 10 levels) was entered as a random part of models, as it could be expected to have a non-systematic or unpredictable influence on our data. All variables were standardised before analysis to enable direct comparison of regression coefficients.

## 3. Results

We gathered 28 fresh and 30 remains samples. We did not obtain amplification for two remains samples, possibly due to excessive DNA degradation. Thus, we have genotyped 28 fresh and 28 remains samples (Appendix A). The amplification success (100% vs. 93.3%) did not differ significantly between the two groups (*Z* = 1.39, *p* = 0.16).

All loci were polymorphic in both types of samples, with the number of alleles ranging between three and five in the combined dataset (Table 1). The number of alleles did not differ between sample types (medians 3 and 3.5 alleles per locus for fresh and remains samples, respectively; *P*(Wilcoxon test) = 0.23). The observed heterozygosity (*H_obs_*) was higher in fresh samples than remains (medians 0.589 and 0.454, *P*(Wilcoxon test) = 0.04) but expected heterozygosity (*H_exp_*) showed no significant differences (medians 0.562 and 0.55; *P*(Wilcoxon test) = 0.92), just as *F_is_* (−0.065 vs. 0.147; *P*(Wilcoxon test) = 0.15).

Estimated population allele frequencies were nearly perfectly correlated for both groups (*r* = 0.97, *p* < 0.0001; Figure 3d). The null hypothesis of the linear hypothesis test of intercept = 0 and slope = 1 confirmed agreement between allele frequencies observed for remnant and fresh samples (*F*_47,45_, *p* = 0.89).

INEst pointed to two loci (CC_01 and CC_06) in both types of samples as potentially affected by the presence of null alleles (Figure 3a). However, the lower limit of the 95% confidence intervals around null allele frequencies included zero for all loci. Additionally, genotyping failure rates (*b_j_*) were not significantly higher than zero for all loci and sample type combinations (Figure 3b).

Individual inbreeding rates (*f_i_*) estimated with INEst after accounting for null-alleles and genotyping failures were biased toward higher values in remains samples (Figure 3c). However, considering the wide 95% confidence intervals, in no case was *f_i_* significantly greater than zero.

Fresh and remains samples were not significantly differentiated (*F_st_* = −0.005, with bootstrap confidence intervals from –0.010 to 0.000), which indicated that birds prey on the available beetles irrespective of their microsatellite alleles. In line with this finding, the STRUCTURE analysis did not distinguish fresh samples from remains, and the most plausible number of clusters (*K*) was equal to one (Figure 3g). Both types of samples exhibited no spatial genetic structure; i.e., pairs of individuals were unrelated, regardless of distance (Figure 3e,f). Thus, we may conclude that a single, near-panmictic population inhabits the whole area or that the population is large enough to prevent related individuals’ mating.

Linear mixed models (LMM) regression showed that sample type had no influence on amplification efficiency (*β* = 0.00, SE = 0.25, *t* = 0.001, *p* = 0.99). The most influential variable was the length of the allele (*β* = −12.60, SE = 0.65, *t* = −19.38, *p* < 0.001); it was also much stronger than the effect of individual heterozygosity (*β* = −0.76, SE = 0.04, *t* = −16.09, *p* < 0.001) (See Figure 4). This result indicates that longer alleles in heterozygotes can drop out and that both types of samples can be affected by this effect.

## 4. Discussion

This study demonstrated the successful amplification of microsatellite markers in *C. cerdo* using dead beetles’ remains as a DNA source, indicating that such a non-invasive sampling approach could provide satisfactory material for population–genetic studies without the need of killing or injuring individuals of this vulnerable species. Of particular importance, we obtained highly correlated allele frequencies between fresh and remains samples, and both types of samples resulted in the convergent description of the population’s genetic variation.

Inaccurate estimation of allele frequencies could have severe consequences for the inference of genetic processes acting in the population of species of conservation concern. Particularly, the loss in heterozygosity due to allelic dropout can generate a false Wahlund effect, suggesting (erroneously) that a substructure exists within the sample [6]. Subsequently, the assignment of individuals to the unfounded population of origin can cause erroneous estimates of migration. In our study, samples collected from a single population were not clustered according to the sample type, indicating that the reduction in observed heterozygosity in non-invasive samples was not large enough to cause a misdiagnosis of the population structure.

On the other hand, our results show that caution should be exercised when inferring population inbreeding rates from lower-quality samples. This is because allelic dropout (which arises due to a tendency of weaker amplification of longer alleles, especially in heterozygotes) can reduce the observed heterozygosity (*H_obs_*), potentially resulting in erroneous concerns about extinction risks caused by inbreeding [6]. Inbreeding is often invoked as a severe genetic threat, reducing fitness when the population is small and/or individuals have low dispersal abilities [46,47]. As shown in our study, the use of body remains in the molecular analysis may lead to an overestimation of inbreeding. It should be assumed that DNA in the remains is more degraded than in fresh samples, which would explain the observed differences between both types of samples. In our study, individual inbreeding estimates tended to be higher for remains samples than for fresh samples, but in no case were they significantly higher than zero, and the difference between fresh and remains samples was not statistically significant. Therefore, we can conclude that the studied population was not at risk of increased inbreeding, especially since the population’s spatial genetic structure indicated unlimited gene flow within the studied area. In this respect, our results differ from the previous studies on Polish populations of *C. cerdo*, which pointed to inbreeding as a significant threat to the population’s persistence [48]. On the opposite, our results are in line with previous publications that reported considerable dispersal abilities and high genetic diversity in central- and south European populations of *C. cerdo* [11,32,33].

In earlier studies, non-invasive insect samples were occasionally found not to be suitable for genetic analyses [8,9,49]. It is worth considering what might be the cause of our success. All the limitations of non-invasive sampling methods result from either low DNA quantity, low DNA quality (i.e., degraded DNA), or poor extract quality (i.e., the presence of PCR inhibitors) [6]. An important factor is the amount of tissue available for extraction. It is much smaller in the case of exuviae or insects’ secretions than in dead insect remains. For this reason, the material we used in this study or traffic-killed beetle specimens collected from the road surface [22] could serve as a better source of high-quality genomic DNA than faecal pellets or shed exuviae [8,9,49]. Additionally, in some insects (for example, dragonflies), exuviae could be collected exclusively in wet places; hence, they are more susceptible to bacterial or fungal decay than beetle debris remaining in moderately moist or dry places.

The inability to determine how long the collected body debris of *C. cerdo* were exposed to decay prevented us from including the age of remains as a potential factor of amplification efficiency. Although chitinous debris can lie under trees for years, we can assume that our samples were relatively fresh, as indicated by the high amplification success rate. Our general recommendation to avoid old and more degraded material is to visit occupied sites as often as possible during the beetles’ flight season, collect the remains regularly, and preserve samples until extraction in 90% ethanol and at a low temperature. Such an approach proved successful in our research. Other studies, which used several types of non-invasive DNA sources, emphasised the need to use fresh samples and their conservation in ethanol [9,50].

Our analysis revealed that the sample type influences PCR products’ quality to a minor degree compared to the amplified allele’s length or heterozygosity of an individual. Therefore, to avoid potential genotyping errors, SSR markers with shorter fragment lengths should be recommended for genotyping remains samples as amplification success decreases with increased fragment length [4].

The obvious limitation of the proposed method is the ease of finding insect remains, and that their distribution must be consistent with the intended sampling design. Both conditions could be met when the species and its predators are relatively abundant. We believe that the proposed methodology is promising also for other insects of conservation concern, especially those that have SSR markers developed. For example, the stag beetle *Lucanus cervus*, similarly to *C. cerdo*, lives in oak forests characterised by the presence of old trees. Imagines of both species appear in a similar period and are often predated by birds [51]. Due to large body size, remains of predation of *L cervus* can easily be found along forest roads and could be used for DNA extraction. Suitable SSR markers have been developed for this species [52].

In our study area, *C. cerdo* was the only representative of its genus. However, in some parts of the Palearctic, the species may co-occur sympatrically with several other *Cerambyx* species. Five species belonging to the genus *Cerambyx* occur together in forest ecosystems of south-central or western European countries: *C. cerdo*, *C. miles*, *C. scopolii* and *C. welensii* [35]. Except for the smallest *C. scopoli*, the remains of these species may be similar to each other and could go unrecognized during collection in the field. Nevertheless, we believe that the proposed method can also be applied to sites where a greater number of congeneric species coexist as they would be easily discriminated using microsatellite data and clustering methods, such as STRUCTURE [40] or similar analyses. Such a method would also allow the identification of possible hybrids, as previous research indicated the potential gene flow between *C. cerdo* and *C. welensii* [53]. Barcode mtDNA sequences can serve as an additional aid for species identification [53].

## 5. Conclusions

We provided evidence that the remains of *C. cerdo* left by bird predators could be used as a non-invasive source of DNA for molecular analysis with SSR markers. Although such samples may be susceptible to genotyping failures due to unknown DNA degradation levels, we found that allele fragment length was a more critical determinant for the dropout of one of the two alleles in heterozygotes than whether the sample was from alive insects or debris. When using potentially degraded samples, we advocate applying statistical methods (i.e., INEst or similar methods) that allow the identification of the genotyping failures, mainly because the most significant risk associated with the use of degraded samples is the possibility of overestimating inbreeding due to allelic dropout.

## Figures and Tables

**Figure 1 insects-12-00574-f001:**
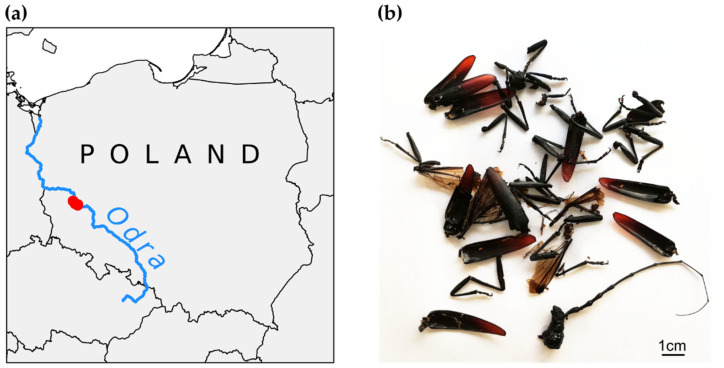
(**a**) Location of study site in Poland. (**b**) Examples of body remains of *Cerambyx cerdo*.

**Figure 2 insects-12-00574-f002:**
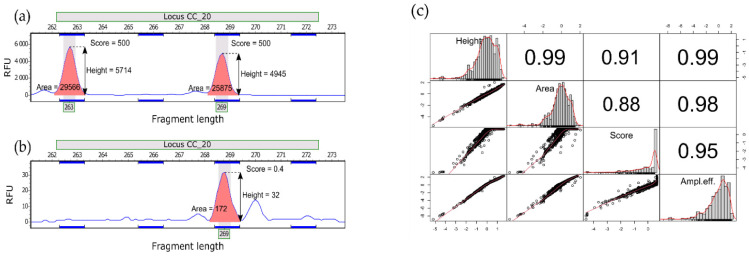
(**a**,**b**) Two electrophoresis profiles showing the peak variables used in this study. (**a**) sample (heterozygote) characterised by higher quality score (**b**) sample (homozygote) with poorer quality peak. The vertical axis is in relative fluorescence units (RFU) and the horizontal axis is in base pairs. (**c**) Correlations between the three peak variables and amplification efficiency—an additional variable computed as the first principal component (PCA) from these three variables to be used as a single dependent variable in regression analysis. All correlations significant at *p* < 0.001.

**Figure 3 insects-12-00574-f003:**
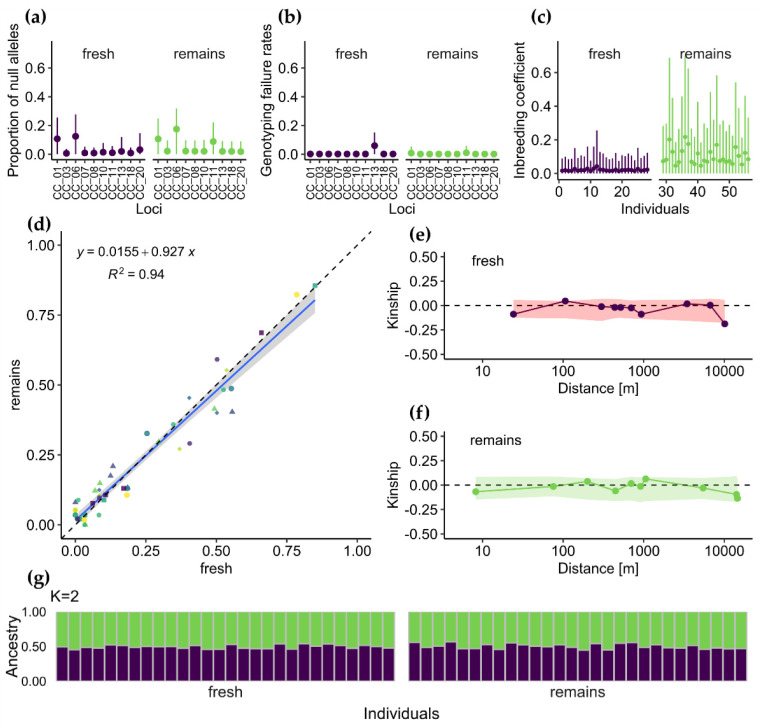
(**a**–**c**) Results from INEst [35] analysis showing (**a**) proportion of null alleles and (**b**) genotyping failure for loci tested, and (**c**) estimated individual inbreeding coefficient, *f_i_*. (**d**) Relationship between allele frequencies in fresh and remains samples (colours and shapes correspond to alleles at different loci). (**e**,**f**) Average kinship coefficients between pairs of individuals plotted against geographical distance for (**e**) fresh and (**f**) remains samples. Points show observed value of pairwise kinship coefficient for mean value of each distance class, while shaded areas around zero represent 95% confidence interval obtained under the null hypothesis that genotypes are randomly distributed. (**g**) Graphical output from STRUCTURE [38]. Each vertical bar represents an individual, and the colour composition displays the probability of belonging to each of the two clusters defined by STRUCTURE.

**Figure 4 insects-12-00574-f004:**
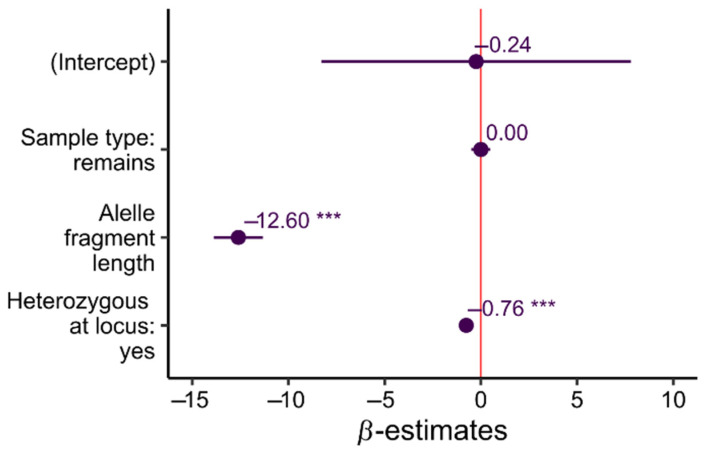
Estimated regression coefficients (β) from the different fixed effects, including the intercept, for linear mixed-effect model. Error bars represent the corresponding 95% confidence intervals. Variables were standardised before analysis to enable direct comparison of coefficients. *** *p* < 0.001.

**Table 1 insects-12-00574-t001:** Population genetic parameters estimated for the two types of samples groups (fresh—samples taken from alive beetles; remains—samples of body remains): *N_Gen_*—number of genotypes, *Miss*—proportion of missing data, *N_All_*—number of alleles, *H_exp_*—expected heterozygosity, *H_obs_*—observed heterozygosity, *F_is_*—inbreeding coefficient.

	Fresh						Remains					
Locus	*N_Gen_*	*Miss*	*N_All_*	*H_obs_*	*H_exp_*	*F_is_*	*N_Gen_*	*Miss*	*N_All_*	*H_obs_*	*H_exp_*	*F_is_*
CC_06	28	0	3	0.25	0.535	0.533	28	0	4	0.250	0.684	0.634
CC_20	28	0	2	0.214	0.300	0.286	28	0	3	0.250	0.229	−0.092
CC_10	28	0	3	0.214	0.198	−0.084	28	0	3	0.179	0.168	−0.063
CC_08	28	0	3	0.643	0.589	−0.092	28	0	4	0.536	0.613	0.126
CC_11	28	0	4	0.571	0.571	−0.001	27	0.036	4	0.407	0.537	0.241
CC_18	28	0	3	0.607	0.557	−0.090	28	0	4	0.500	0.573	0.128
CC_07	28	0	3	0.607	0.567	−0.070	28	0	3	0.500	0.600	0.166
CC_01	28	0	3	0.214	0.411	0.479	27	0.036	3	0.259	0.352	0.263
CC_03	28	0	3	0.679	0.566	−0.199	28	0	3	0.500	0.530	0.056
CC_13	26	0.071	5	0.654	0.617	−0.060	28	0	4	0.571	0.691	0.173
Medians	28	0	3.0	0.589	0.561	−0.065	28	0	3.5	0.454	0.555	0.147

## Data Availability

Data are included in the Appendix A.

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
