# Peer review of "Body Remains Left by Bird Predators as a Reliable Source for Population Genetic Studies in the Great Capricorn Beetle Cerambyx cerdo, a Veteran Oak Specialist"

_insects, 2021, doi:10.3390/insects12070574_

Round 1
Reviewer 1 Report
Non-invasive genetic sampling increases safety for both animal subjects and researchers, minimizes disturbance to animals. This is a relatively straightforward paper demonstrating that non-invasive remains samples could provide satisfactory data for population-genetic studies in Cerambyx cerdo. The authors claim that the same methodology is promising also for other insects of conservation concern, however, it would not be feasible for the animals, which have not SSR markers developed, as developing SSR markers is pretty time and cost consuming. Except for measuring purity and concentration, there are some more efficient techniques checking DNA quality, such as Bioanalyzer and pulsed-field gel electrophoresis. Additional optimized protocols and a thorough discussion about the implementation of those protocols are therefore necessary before the study would be of interest to readers of insects as well as usage in other insect species.
Author Response
The authors claim that the same methodology is promising also for other insects of conservation concern, however, it would not be feasible for the animals, which have not SSR markers developed, as developing SSR markers is pretty time and cost consuming.
Response: We thank the Reviewer for this helpful suggestion. We added relevant information in L350-352 (“ The same methodology is promising also for other insects of conservation concern, especially those which have SSR markers developed.”)
Except for measuring purity and concentration, there are some more efficient techniques checking DNA quality, such as Bioanalyzer and pulsed-field gel electrophoresis. Additional optimized protocols and a thorough discussion about the implementation of those protocols are therefore necessary before the study would be of interest to readers of insects as well as usage in other insect species.
Response: We would like to thank the Reviewer for pointing out additional DNA quality testing techniques. However, the problem is that the extract must not only be of high quality in terms of purity and concentration but also be obtained from the genome of the target organism. All mentioned methods are of little help in the case of DNA extracted from forensic-type material exposed to bacterial or fungal decay. Usually, a very high concentration of DNA can be isolated from the remains of dead specimens, and thanks to advanced extraction methods, it is possible to obtain high purity of the extract. Unfortunately, most of it will be non-specific DNA, i.e. from organisms that decompose the debris and not from the target organism. After all, there is probably no better way to check whether a DNA extract is suitable for amplifying a specific marker than to test this possibility in practice (as we did).
Reviewer 2 Report
In the MS, the authors have tried DNA extraction from non-invasive sampling using the body remains left by bird predators and then analyzed population genetics by microsatellite marker using DNA extraction from them. Although I have no practical experience of the methods in the MS, both the DNA extraction experiments and population genetic studies seem to have been competently conducted. The submitted MS is considered suitable for publication in Insects.
Author Response
We thank the Reviewer for appreciating our work and supporting it for publication in Insects.
Reviewer 3 Report
This is an interesting paper studying the potential use of remains of Cerambyx cerdo predated by birds for the study of the population genetics of the longhorn beetle. Conclusions and scope are interesting although I think that somewhat limited to some forest contexts. My major remarks/comments/suggestions are in the attached file

Author Response
This is an interesting paper studying the potential use of remains of Cerambyx cerdo predated by birds for the study of the population genetics of the longhorn beetle. Conclusions and scope are interesting although I think that somewhat limited to some forest contexts. My major remarks/comments/suggestions are in the attached file
Response: We thank the Reviewer for appreciating our work and supporting it for publication in Insects.
L140-142: What was the exact composition of the trap bait? Did the bait have salt as a preservative? The adults were collected drowned in the bait? If so, how long did they stay inside the bait? As all the aforementioned could theoretically affect the preservation of DNA (although apparently not in practice, line 222), I suggest that the authors clarify this issue a little more in the text. I also suggest adding here one of the available references on the traps used
Response: The trap was filled with red wine only. We used the cheapest wine available at a local store (a five-litre cardboard box; we do not remember a particular brand name, but we think it is of minor importance). After two or three days, the wine in the trap turned sour and became more attractive to insects. Thanks to the installation of perforated partition, the beetles did not drown in the lure and were released after tarsi amputation. So, in the case of fresh samples, we believe that there was no problem with DNA degradation (samples were immediately preserved in ethanol and transported in a portable coolbox to the refrigerator as soon as possible). We added some additional information regarding trapping and trap design and citations of papers describing traps (L140-142).
L178: It is not clear to me how the sampling is structured to apply the Wilcoxon signed-rank tests for paired data. The replicates are true paired samples of both live insects and remains taken both in different stands throughout the study site? Note that otherwise they would be unpaired data. Please explain.
Response: With the Wilcoxon signed-rank test, we compared results obtained for sample types (fresh or remains). The way these data are paired is well shown in Table 1 (fresh vs. remains, pairs by locus).
L240: There is an erratum here, I guess it is "slope = 1"
Response: Indeed, it should be "slope = 1". Corrected.
143, 288 The authors reasonability assume that the birds prey on the available beetles at random (ie they do not differentiate haplotypes). I agree that this is probably the case, but cannot be sure (for example if a certain haplotype is linked to a more elusive behavior of the insect) This (unrandom predation by birds) would theoretically be another potential cause of haplotype dropout. Please consider it
Response: Fresh and remains samples were not significantly differentiated (Fst = -0.005, ns), which may indicate that birds prey on the available beetles irrespective of their microsatellite alleles. We added such information in L 256-257.
Discussion: Although it is perhaps a truism, the authors should indicate somewhere in the discussion that the method proposed is only applicable if in the habitat studied the population of crows (or other species of predatory birds) reaches a minimum size and a spatial distribution compatible with the sampling protocol.
Response: We added such remarks in the paragraph before the last paragraph of the Discussion (L 350-352).
Discussion: I wonder what would happen if C welensii (or another large species of Cerambyx) coexists sympatrically with C cerdo in the studied habitat. Do the markers used allow to differentiate C welensii from C cerdo? (I suppose so, but it is important to stress this point); Could the presence of C welensii be a practical (field or lab) limitation for the proposed method in other areas or countries? I suggest that the authors include a small paragraph in the discussion on this important ecological/methodological aspect. I also think that the MS could benefit from the content of the recent paper by Torres-Vila, L. M., & Bonal, R. (2019). DNA barcoding of large oak-living cerambycids: diagnostic tool, phylogenetic insights and natural hybridization between Cerambyx cerdo and Cerambyx welensii (Coleoptera: Cerambycidae). Bulletin of entomological research, 109(5), 583-594.
Response: We would like to thank the Reviewer for pointing to an interesting publication. It could be expected that if other Cerambyx species exist sympatrically with C. cerdo in the studied habitat (which was not the case in our study), they would be easily differentiated using microsatellite data and STRUCTURE analysis (or a similar method). Such a method would also allow the identification of possible hybrids. We added an additional paragraph on these aspects at the end of the Discussion (L 360-371).
Reviewer 4 Report
The submitted manuscript by Redlarski et al is discussing the use of body remain for population genetic studies in Cerambyx cerdo beetle. Overall, it highlights the impact of that methodology for population genetics studies in that threatened species which could be applied into other insects of conservation concern, too. In general, the manuscript is well-written while I have some comments to suggest to the authors. Please find below my comments in detail.
Materials and methods as well as results would be better broken down into sections with specific titles for better understanding.
Figure 1a: Difficult to distinguish the 2 dots on the photo indicating the collection site. Maybe add a smaller figure in higher magnification.
Figure 2: The graphs are very small so it's difficult to read them. Please, re-adjust the scale.
Results: Was the quantity and quality of DNA extractions evaluated by Nanodrop? If yes, it could be included as a table at the section of results.
Line 221: Were the 2 remaining samples excluded randomly from the analysis?
Figure 3 g: Increase the font size.
In general, I would suggest the size of the figures as well as the font size of the included text to be increased.
Line 317: "In earlier studies" , references need to be included.
Author Response
The submitted manuscript by Redlarski et al is discussing the use of body remain for population genetic studies in Cerambyx cerdo beetle. Overall, it highlights the impact of that methodology for population genetics studies in that threatened species which could be applied into other insects of conservation concern, too. In general, the manuscript is well-written while I have some comments to suggest to the authors. Please find below my comments in detail.
Response: We thank the Reviewer for his/her careful reading of the manuscript and constructive remarks.
Materials and methods as well as results would be better broken down into sections with specific titles for better understanding.
Response: Perhaps it would be a good idea if the sections were longer. In our text, the Results and Methods are so concise that creating sections with 2-3 sentences each would be redundant, in our opinion.
Figure 1a: Difficult to distinguish the 2 dots on the photo indicating the collection site. Maybe add a smaller figure in higher magnification.
Response: Our intention was not to indicate two dots but a section along the river. We would not like to overcomplicate the illustration, but only indicate the approximate location of the study site.
Figure 2: The graphs are very small so it's difficult to read them. Please, re-adjust the scale.
Response: We propose to enlarge the figure by fitting it to the entire width of the page.
Results: Was the quantity and quality of DNA extractions evaluated by Nanodrop? If yes, it could be included as a table at the section of results.
Response: Yes, we evaluated DNA concentration while the extracts were diluted for PCR (we are not sure if we would still be able to find notes with measurement values). We decided not to use concentration as an additional variable as we found it to be of little use in the case of DNA extracted from forensic-type material exposed to bacterial or fungal decay. The problem is that the DNA extract must not only be of high quality in terms of purity and concentration but also be obtained from the genome of the target organism. A very high concentration of DNA can be isolated from the remains of dead specimens, and thanks to advanced extraction methods, it is possible to obtain high purity of the extract. Unfortunately, most of it will be non-specific DNA, i.e. DNA from organisms that decompose the debris and not from the target organism. After all, there is probably no better way to check whether a DNA extract is suitable for amplifying a specific marker than to test this possibility in practice (as we did).
Line 221: Were the 2 remaining samples excluded randomly from the analysis?
Response: These two samples failed, possibly due to excessive DNA degradation (information added in L 226).
Figure 3 g: Increase the font size.
Response: We increased the font size in Figure 3 g.
In general, I would suggest the size of the figures as well as the font size of the included text to be increased.
Response: We increased size of Figure 4. Other figures can be enlarged by fitting them to the entire width of the page, but the decision to enlarge them is left to the editorial office.
Line 317: "In earlier studies" , references need to be included.
Response: References included (L 324).